# A combined Terra/Aqua MODIS snow-cover and RGI6.0 glacier product (MOYDGL06*) for the High Mountain Asia between 2002 and 2018

Sher Muhammad[1, 2], Amrit Thapa[1]

[1]International Center for Integrated Mountain Development (ICIMOD), Kathmandu, Nepal
[2]Institute of International Rivers and Eco-security, Yunnan University, 650500 Kunming, China

*Correspondence to*: Sher Muhammad (sher.muhammad@icimod.org)

**Abstract.** Snow is a significant component of the ecosystem and water resources in the High Mountain Asia (HMA). Accurate, continuous and long-term snow monitoring is necessary for water resources management and economic development. In this study, we improved Moderate-resolution Imaging Spectroradiometer (MODIS) onboard Terra and Aqua snow–cover for HMA by a multi-step approach. The primary purpose of this study was to reduce uncertainty in MODIS snow cover. For reducing underestimation mainly caused by cloud cover, we used seasonal, temporal, and spatial filters. For reducing overestimation caused by MODIS sensor, we combined MODIS Terra and Aqua snow-cover products considering snow only if a pixel is snow in both the products otherwise no snow, unlike some previous studies considering snow if any of the Terra or Aqua product is snow. Our methodology generates a new product which removes a significant amount of uncertainty in raw MODIS 8-day composite product comprising 46% overestimation and 3.66% underestimation, mainly caused by sensor limitations and cloud cover, respectively. The results were validated using Landsat 8 data as ground truth, both for winter and summer at twenty well-distributed sites in the study area. Our validation results show that the adopted methodology improved accuracy on average by 10%, mainly reducing the snow overestimation. The final product covers the period from 2002 to 2018, as a combination of snow and glaciers created by merging RGI6.0 glacier boundaries separately debris-covered and debris-free to the final snow product namely MOYDGL06*. Each of the Terra and Aqua datasets contains seven hundred and forty-six image files derived initially from approximately one hundred thousand satellite individual images. The data is available for researchers to use for various climate and water-related studies. The data is available at https://doi.pangaea.de/10.1594/PANGAEA.901821 (Muhammad and Thapa, 2019).

## 1 Introduction

Snow is a crucial component of the hydrological cycle, acts as water storage with a short delay in the seasonal runoff (Colbeck, 1977). More than 60% of the annual discharge in the major rivers of High Mountain Asia (HMA) depend on melt-water on average with variable rates across the region (Armstrong et al., 2018). Both the mountain communities and downstream population rely on water stored as snow for their daily use mainly in the early melt-season (Lutz et al., 2016). On the contrary, rapid snowmelt may cause natural hazards such as floods, consequently damage agriculture, infrastructure, and human life (Haq et al., 2012; Memon et al., 2015). These factors make it essential to monitor snow for downstream water resources management and hazards/disasters preparedness (Clifton et al., 2018; Tian et al., 2017; Zhang et al., 2010).

Snow cover mapping is generally crucial for areas densely populated downstream, and where snowmelt dominates the discharge (Smith et al., 2017). In the topographically complex High Mountains of Asia snow covers a vast spatial extent which is difficult to measure in the field (Immerzeel et al., 2009). Hence, cryospheric field observations are limited to the lower elevation zones with less spatial coverage (Muhammad et al., 2019a, 2019b; Muhammad and Tian, 2016). Field data from very few weather stations are available with limited regional coverage. These direct observations do not provide representative samples of the snow conditions globally and in the region (Latif et al., 2019; Möller and Möller, 2019; Wunderle et al., 2016).





Therefore, remote sensing data is mostly used to assess the snow extent and variability at regional or global scales (Hall et al., 2010).

Satellite data provide broad coverage and is capable of continuous long-term monitoring of snow since recent half-century (Hüsler et al., 2014). The primary constraint in passive satellite remote sensing is the cloud persistence for regular

spatiotemporal monitoring of various earth resources including snow (McCabe et al., 2017). Due to this fact, 8-day composite snow cover products derived from Moderate-resolution Imaging Spectroradiometer (MODIS) were developed to minimise the persisting cloud cover over the snow (Hall et al., 2002). Although the 8-day composite product reduced the cloud cover, still a significant amount of clouds remained particularly in the monsoon and winter precipitation season (Liang et al., 2008). The presence of clouds may underestimate the snow cover extent and must be removed (Wang et al., 2008). Also, obscuration of

old snow and glacier ice due to their low albedo are challenging for MODIS to capture and are the contributing factors in underestimation of snow and ice cover extent. In contrast, the larger Sensor Zenith Angle (SZA) (Li et al., 2016) and low spatial resolution (Hou et al., 2019; Huang et al., 2017) mainly causes overestimation of snow. The overestimation is also significantly influenced by the broad swath of MODIS that amplifies the edge-pixels more than four times compared to the pixels at the image centre (Zeng et al., 2011; Zhang et al., 2017). Further, MODIS tends to overestimate snow cover in the

evergreen forests and the early melt season (Hall and Riggs, 2007).

Several studies tried to improve the snow cover extent to reduce uncertainty. Gurung et al., (2011) estimated seasonal snow cover in the HKH region combining Aqua and Terra satellites followed by temporal, spatial filter and altitude mask mainly to minimise the cloud cover. Hammond et al., (2018) generated global snow zone maps and calculated trends in snow persistence using Terra product and reduced the overestimation by excluding snow persistence (SP) to less than 7 %. Basang et al., (2017)

analysed the snow cover in Tibet using Terra satellite and ground observation, concluding that combining remote sensing data with ground observations reduces the uncertainty. Although these studies somehow improved the quality of snow cover, the data require further improvement to reduce the remaining error of commission and omission (Riggs et al., 2016). This study not only removes cloud persistence causing underestimation but reduces an enormous amount of overestimation in snow cover caused by MODIS sensor. A long-term (2002-2018) meticulous estimate of snow cover for the HMA (Fig. 1) will facilitate

climate, glacio-hydrological modelling, understanding the present dynamics of the cryosphere in the region (Brun et al., 2017; Muhammad et al., 2019a), and develop associated products, e.g., snow water equivalent (Alonso-González et al., 2018; Painter et al., 2016).

## 2 Data

MODIS sensor is onboard the Terra and Aqua satellites of NASA launched in 1999 and 2002, respectively. It provides land

surface and cloud data in 36 spectral bands within 0.4 to 14.4 mm of the electromagnetic spectrum. The local equatorial pass time of Terra is 10:30 a.m. in descending node and for Aqua 01:30 p.m. in the ascending node. Snow cover is one of the widely used products of MODIS, available through the website www.nsidc.org of the National Snow and Ice Data Center (NSIDC) and https://earthdata.nasa.gov/ of NASA's Earth Science Data Systems (ESDS). The snow product is available at 500 m and 5 km spatial resolution with daily, eight days, and monthly temporal resolution. This study uses 8-day maximum snow extent

product version 6 of the MODIS onboard Terra (MOD10A2.006*) and Aqua (MYD10A2.006*) available from February 2000 and July 2002, respectively with 500 m spatial resolution for the Hindukush, Karakoram, and Himalaya (HKH) and surroundings. This version minimises the error of omission and commission compared to version 5 primarily in clear sky conditions as described by Riggs et al., (2016). In collection 6, band 6 of AQUA is restored instead of the previously used band 7 in calculating NDSI making the algorithm similar to that used for TERRA (Riggs et al., 2016) which helps to reduce

an additional uncertainty in AQUA snow cover. The 8-day composite product depicts snow if it is observed in any of the eight days either once or multiple times. The data are classified as 0 (missing data), 1 (no decision), 11 (night), 25 (no snow), 37



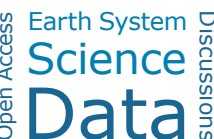

(lake), 39 (ocean), 50 (cloud), 100 (lake ice), 200 (snow), 254 (detector saturated), and 255 (fill) (Riggs et al., 2016). One MODIS tile is approximately $1200 \times 1200$ km ($10° \times 10°$) swath. We used Landsat 8 data with 30 m spatial resolution as ground truth to validate the MODIS snow cover. We used a total of 20 Landsat scenes (10 for peak snow cover and 10 for minimum snow cover period) of the year 2018.

## 3 Method

One of the major issues in the passive remote sensing data is the cloud cover which is more prominent in the mountainous regions. The existence of cloud cover was the primary reason for developing the 8-day composite snow cover product, produced by merging eight consecutive days of MODIS images (Hall et al., 2002). A significant amount of clouds remains in the 8-day composite product causing underestimation in the snow cover extent and needs to be removed for making the product useful for various climatological and glacio-hydrological applications (Yu et al., 2016). In addition, the overestimation was removed by combining Aqua and Terra to estimate snow with more confidence. We used a multi-step approach to remove all the clouds and make a combined Terra/Aqua snow cover cloud-free product for the High Mountain Asia for the period of 2002 to 2018. The detailed methodology contains the following steps applied separately to both Terra (MOD10A2.006*) and Aqua (MYD10A2.006*) followed by combining them. The methodology is also described as a flow chart in Figure 2.

### 3.1 Seasonal filter

We converted the data into the snow and no snow followed by classifying all the images into two seasons by selecting data from 15th April to 15th October as representative of summer and the rest as in winter season of a hydrological year. Then for each year, each season ͦs data was merged, and the total seasonal accumulated snow cover extent was used to extract the raw data, to remove the cloud beyond the maximum snow extent. We call the original MOD10A2.006 and MYD10A2.006 as raw data throughout the manuscript. The data (cloudy pixels) beyond the maximum snow extent were converted to no snow for further processing. This step was performed to reduce the time consumption for the next steps and possible uncertainty in removing cloud cover by temporal and spatial filters.

### 3.2 Temporal filtering

The clouds after the seasonal filter were removed by applying a temporal filter. This filter replaces the cloudy pixel by non-cloudy pixels from the chronological preceding and subsequent images (Gao et al., 2010; Hüsler et al., 2014; Li et al., 2019b; Paudel and Andersen, 2011; Zhang et al., 2017). The length of the temporal filter window should be carefully considered. A 7–day temporal filter applied to the daily MODIS data reduced more than 95% of the cloud cover over Austria (Parajka and Blöschl, 2008). Tran et al., (2019) used a 30-day-period for the temporal filter to remove long-lasting clouds. In this study, after testing several images, we selected four images (two preceding and two subsequent 8-day composite images) at most for removing cloudy pixels. For each cloudy pixel, the same pixel in the following image was checked. If the pixel is snow or no snow then cloudy pixel was replaced accordingly; otherwise, the previous image was tested with similar criteria. The criteria are continued up-to two preceding and following images in case of cloud persistence. If the clouds remain continuously in all four images, then we go for the spatial filter. For the temporal filter, we assumed that the snow cover remained constant under continuous cloudy conditions (Gafurov and Bárdossy, 2009). However, this assumption may not work in case of possible melting which is expected to be negligible. Following Eq. (1-3) explain the temporal filter. These equations are stepwise; if the condition is satisfied in the first step, then the other steps are not followed, and the filter goes to the next pixel to check the conditions. The equations convert cloud to no snow if the snow is no snow in the following equations. The condition of snow to no snow is satisfied in Eq. (1) only replacing the "OR" as "AND". Conversely, if all the conditions in equation (3) are cloud, then the pixels remain cloudy in the temporal filter and is considered for conversion to snow or no snow by the spatial filter.





Step 1: $S^C_{(y,x,t)} = snow$ **IF** $\left( S_{(y,x,t-1)} = snow \textbf{ OR } S_{(y,x,t+1)} = snow \right)$ (1)

Step 2: $S^C_{(y,x,t)} = snow$ **IF** $\left( S_{(y,x,t-1)} \textbf{ AND } S_{(y,x,t+1)} = cloud \textbf{ AND } S_{(y,x,t-2)} = snow \right)$ (2)

Step 3: $S^C_{(y,x,t)} = snow$ **IF** $\left( S_{(y,x,t-1)} \textbf{ AND } S_{(y,x,t+1)} \textbf{ AND } S_{(y,x,t-2)} = cloud \textbf{ AND } S_{(y,x,t+2)} = snow \right)$ (3)

### 3.3 Spatial filtering

The majority neighbourhood spatial filter was applied to the remaining cloudy pixels in the images after the temporal filter. We used this filter after the temporal filter (which removes the majority of the clouds) and because the spatial filter is useful for small/patchy clouds (Li et al., 2019a). It reclassifies the cloudy pixel to snow or no snow based on the majority of the non-cloudy surrounding (eight neighbouring) pixels in a 3*3 window (Parajka and Blöschl, 2008). When there is a tie between no snow and snow pixels in the surroundings, the particular pixel is assigned as snow. Also, running this filter does not remove all the remaining cloudy pixels when applied only once. The pixels remain cloudy only when all the eight neighbourhood pixels are cloudy. The criteria of the spatial filter are also described in figure 3.

### 3.4 Combine Terra and Aqua data

After filtering, we found that both the datasets are overestimating snow, particularly at a lower elevation. We assumed that the approximate three hours difference in an acquisition time of Terra and Aqua do not affect the snow conditions (snowfall/snowmelt). Earlier studies combined Terra and Aqua, assuming snow if the pixel is snow in any of the images (Parajka and Blöschl, 2008; She et al., 2015; Xie et al., 2009; Yu et al., 2016). We combined both Terra and Aqua in a way by considering snow only where pixels in both the products are classified as snow. The criterion is also an inter-verification of snow mapped by Terra and Aqua. It also helps us to avoid uncertainty produced using the cloud removal methodology as described in section 3.3 by any of the Terra or Aqua data. This step significantly improves the snow product, mainly reducing the overestimation in the images captured from off-nadir view (Li et al., 2016) and edge-pixels replication due to the broad swath of MODIS (Zeng et al., 2011; Zhang et al., 2017). The cloud cover removed in all the images during the study period by the methodology described from section 3.1 to 3.4 for both Terra and Aqua is shown in Figures 4 and 5. The data of both Aqua and Terra overlap from late 2002; therefore, the 8-day composite product was generated from 2002 to 2018 in this study. The method of combining snow from Terra and Aqua is described in Eq. (4-5). We do not recommend this method for daily snow product in mountainous areas because the error of omission may be further increased because of the off-nadir view acquisition/edge-pixels.

Step 1: $S^{Combined}_{(y,x,t)} = snow$ **IF** $\left( \left( S^{T_{final}}_{(y,x,t)} = snow \textbf{ OR } cloud \right) \textbf{ AND } \left( S^{A_{final}}_{(y,x,t)} = snow \right) \right)$ (4)

Step 2: $S^{Combined}_{(y,x,t)} = snow$ **IF** $\left( \left( S^{T_{final}}_{(y,x,t)} = snow \right) \textbf{ AND } \left( S^{A_{final}}_{(y,x,t)} = snow \textbf{ OR } cloud \right) \right)$ (5)

where $T_{final}$ and $A_{final}$ are Terra and Aqua final products, respectively.

### 3.5 Combine glaciers (RGI6.0) to the snow product

In the regions where snow and glaciers both exist, it is challenging to differentiate particularly in the accumulation period. Also, the glacier ice mainly in the late ablation season is difficult to map using the MODIS algorithm for snow detection when the albedo of the glacier surface is comparatively low. MODIS is incapable of mapping ice under the debris. Therefore, we used the latest Randolph Glacier Inventory version 6.0 (RGI6.0) (RGI Consortium, 2017), partly developed by Mölg et al., (2018) and supraglacial debris cover for RGI 6.0 by Scherler et al. (2018), resampled into the MODIS pixel size and merged



it into the combined MODIS data. A combined snow and glacier cover (debris-covered and debris-free) product was developed which will be useful mainly for glacio-hydrological applications.

### 3.6 Data description – Product coding

The final MODIS Terra (MOD10A2) and Aqua (MYD10A2) version 6 combined with Randolph Glacier Inventory version 6

(RGI6.0) were named as MOYDGL06. In the final product, we flagged the pixels which were changed from raw product either from no snow to snow, or the other way around. The values in the final product were classified as 0 if no snow, 200 if the pixel is snow in the raw and final product and −200 if snow is converted to no snow in the final product. If no snow is converted to snow mainly under cloud cover, the value is flagged as 210, exposed debris-covered and debris-free ice are numbered as 240 and 250, respectively. The glacier ice (debris-covered and debris-free) shielded by snow is classified as snow and flagged as

200. All the improved snow data of the combined product throughout the study period is shown in Figure 6. The combined product will especially be useful for many hydro-glaciological applications. If only snow data is required, then the values −200, 240, and 250 be considered as no snow while 200, and 210 represent the improved snow.

### 3.7 Validation of the product using Landsat data

The final product was validated to assess the accuracy of the improved snow product using snow derived from Landsat 8

images as ground truth for the year 2018 during both summer and winter seasons. The snow was classified in Landsat following similar criteria applied for MODIS snow product, using NDSI based on Landsat bands 3 (0.53–0.59 μm) and 6 (1.57–1.65 μm) followed by the reflection in near-infrared light greater than 11 % to prevent water from being incorrectly classified as snow. MODIS data were resampled to the Landsat pixel resolution before comparison. A well-distributed twenty Landsat scenes throughout the study area were compared to the combined Terra and Aqua snow product to validate our results as shown

in Figure 1. We selected cloud-free (<5%) Landsat images except for one site (Nepal) where the clouds were approximately 7% due to persistent cloud cover throughout the year. The overall accuracy of the Terra/Aqua, raw and processed and their combined final product is shown in Tables 1 and 2. The overall accuracy is not necessarily improved in all the cases mainly due to the cloud cover and the overestimation of snow by MODIS in raw data. The combined product shows a significant improvement over the raw snow data as compared to Landsat data shown in Figure 7.

## 4 Results and discussion

This study generated a combined Terra and Aqua 8-day composite snow together with glacier (debris cover, debris free) product namely MOYDGL06 for the period between 2002 and 2018. The study period started from the year 2002 as Aqua satellite data is available since 2002. We did not use MODIS snow data of the year 2000 in our final product, but it is worthy of highlighting that the snow data till December 10, 2000, contains data voids/strips and are not recommended for any

applications/ analysis. We use existing techniques for cloud removal in addition to uniquely combining Terra and Aqua snow to predominantly reduce the overestimation. The first step (seasonal filter) removed approximately 44.66 % and 31.29 % of the total cloud cover existing mainly outside the snow cover extent in Terra and Aqua products, respectively. This step does not affect snow data as if there is snow on any day of the half year period; the data in raw data is extracted based on the mask in this step. The second step (temporal filter) removed 54.08 % and 65.48 % of the total clouds which is equal to 98.74 % and

96.77 % of the total removed clouds in combination to the seasonal filter applied on Terra and Aqua snow products, respectively. Temporal filter was the most effective step in cloud removal. The third step (majority neighbourhood spatial filter) removed 99.91 % and 99.84 % of the total clouds in which 1.17 % and 3.07 % were removed itself by the spatial filter in Terra and Aqua snow products, respectively. The spatial filter removes a significant amount of cloudy pixels with minor errors (Paudel and Andersen, 2011). The fourth step of combining Terra and Aqua products also helped to remove 0.06 % and



0.14 % of the clouds in making the product 99.98% cloud-free on average. As a whole, on average, approximately 0.02 % of the total clouds remained in our final product. Our data is available at https://doi.pangaea.de/10.1594/PANGAEA.901821 (Muhammad and Thapa, 2019).

The method of combining Terra and Aqua is also an inter-verification of the snow derived by both the satellites. Our results
indicate that on average approximately 46% of the total snow on average is overestimated by MODIS. This significant difference in the snow data is mainly due to the large swath and low spatial resolution of MODIS which makes it challenging to map snow cover accurately, particularly at the edges of each image. Similarly, the off-nadir view makes the sensor zenith angle larger causing it to replicate the edge pixels. Whereas, the underestimation is mainly caused by the cloud cover but is insignificant, i.e. 3.66% of the snow on average. We are more confident about the MODIS snow cover derived from our
method. Combining the snow with the glacier cover (debris-covered and debris-free) makes it more comprehensive and usable for various hydro-glaciological applications. The glacier ice captured by MODIS as snow is represented as 200 (snow). We combined glaciers uncaptured as snow by MODIS in the combined product representing debris-covered and debris-free ice as 240 and 250, respectively. These values (240 and 250) may be ignored or converted to no snow if the user of the data is interested only in the MODIS snow product. In this case, the values 200 and 210 can be considered as the final snow.

Comparison of the snow cover area estimated by Landsat and MODIS Aqua/Terra raw/final and combined product shows that our methodology improved the accuracy by 10% from 77% to 87% on average reducing the inevitable overestimation for twenty well-distributed (in space and time) Landsat scenes. The remaining overestimation may either require improvement in snow detection algorithm or be constrained by low spatial resolution and large swath. The overall accuracy is incapable of capturing an approximately 46% of the overestimated snow (Figure 6) facilitated by our methodology of combining Terra and
Aqua. The overestimation in Terra and Aqua MODIS 8-day raw products (MOD10A2*/MYD10A2*) is enormous and may not be suitable for statistical analysis and other hydrological applications without improvement. Whereas, on average 3.66% of the snow which MODIS was not able to catch due to cloud cover, our filtering techniques facilitated to convert it into the snow.

It is essential to highlight that the snow persistence threshold as suggested by Hammond et al., (2018) is useful to remove
overestimated snow at low altitudes. At the same time, it can also underestimate snow in some areas particularly in the Tibetan Plateau and in the eastern Himalaya. Although it worked well in the Karakoram and surrounding areas, the inconsistency throughout the region makes this algorithm ineffective, for large scale studies. An example of snow underestimation by 7% persistence threshold is shown in Figure 8. Similarly, some studies used snow line approach to remove overestimated snow at low altitudes and convert cloudy pixels to snow or no snow (Dietz et al., 2013; Krajčí et al., 2014, 2016; Parajka et al., 2010).
However, the use of snow line approach is questionable in complex terrain due to higher elevation variability. As an alternative to both these methods, we recommend using a combination of Terra and Aqua considering snow only if both the satellite map the pixels as snow, otherwise no snow. This criterion removed approximately 46% of the overestimated snow including most of the low altitudes snow, but the overall accuracy is incapable of representing such an enormous enhancement; somewhat it may negatively affect the overall accuracy. An example of the improved snow based on the criteria is shown in Figure 9. Our
accuracy assessment based on Landsat data shows that the snow cover in our final combined snow product is improved approximately by 10% on average as compared to the raw Terra and Aqua snow data. The slight improvement in overall accuracy in the final product is expected mainly because of the MODIS data resolution (Gao et al., 2010; Parajka and Blöschl, 2008). This improvement is mainly due to the cloud removal and conversion of masked snow by clouds to snow. The significantly considerable uncertainty of underestimation is mainly due to cloud cover and overestimation by MODIS data
making the raw MODIS product approximately 50% uncertain which limits the data quality to quantify the snow dynamics without improvement. The raw and final snow time series of 2002-2018 for the whole study area is shown in Figure 10. The raw and improved data products show a significant difference throughout the observation period. The improved data also



include the snow below the cloud cover. Bias in both the data sets is slightly reduced by the snow converted from no snow, mainly due to cloud cover.

Two images were missing in the raw MODIS data that are 2008145 and 2016049 in the time series. To fill the gap, we use the previous images which are 2008137 and 2016041 as a replacement of the missing data. This replacement is based on the
assumption that the snow cover remained the same as in the previous 8-day composite image. As the time series of more than sixteen years is quite large, the replacement of only two missing images will not affect statistical analysis and its use for various hydro-glaciological applications. The overall snow extent shows a significantly decreasing trend since 2013 as compared to the whole observation period between 2002 and 2018. The snow cover in the first decade of the twenty-first century showed an increasing trend, the similar and short observation period was covered by most of the glacier mass balance studies (Brun et
al., 2017; Gardelle et al., 2013; Gardner et al., 2013; Kääb et al., 2012, 2015; Muhammad et al., 2019a, 2019b; Muhammad and Tian, 2016). It might be interesting to estimate and understand the contemporary glacier mass balance and its hydrological impact across in the region.

## 5 Data availability

The data is available at [https://doi.pangaea.de/10.1594/PANGAEA.901821](https://doi.pangaea.de/10.1594/PANGAEA.901821) (Muhammad and Thapa, 2019). A source-code
named "R Code for MODIS filtering and combine.zip" is attached as a supplement. The code comprises a temporal filter, spatial filter, and combining MODIS Terra and Aqua products. The accompanying "Instructions" file in a zip folder gives the necessary information about the prerequisites and how to run to code.

## 6 Conclusion

A combined snow product derived from Terra and Aqua MODIS version 6 and glacier (RGI6.0) named as MOYDGL06* was
developed from 2002 to 2018 covering the High Mountains of Asia. The product consists of the original snow data and pixels, changed from snow to no snow and vice versa, based on our methodology. The value −200 is the overestimated snow by either Terra or Aqua and was converted to no snow by combining Terra and Aqua, 200 is snow in both Terra and Aqua without any change in the final product, 210 is the no snow to snow converted mainly from clouds over snow, 240 and 250 represent debris-covered and debris-free ice, respectively. On average the value −200 is approximately 46% of the original snow (both Terra
and Aqua) for the whole region during the study period whereas, 210 is 3.66% on average mainly due to cloud cover, suggesting that the raw MODIS data is 50% uncertain in comparison to our final combined snow product. On the contrary, we do not recommend combining daily Terra and Aqua snow products as the large SZA may significantly underestimate snow. We conclude that clouds are not the main obstacle in the MODIS 8-day composite product. Our correlation of accuracy assessment shows that our final MODIS product in comparison to twenty well-distributed Landsat scenes improved the accuracy by 10%
from 77% to 87% on average. The hindrance in MODIS data quality is due to the broad swath and low spatial resolution which mainly affect snow conditions in the topographically complex mountainous regions. We believe that the product will be a crucial component for the glacio-hydrological studies in the region and will significantly improve cryosphere monitoring and associated changes.

**Author contributions.**

SM designed the study and developed the methodology. SM and AT applied the methodology. Both the authors contributed to the writing of the manuscript and data quality control.



**Competing interests.**

The authors declare no conflict of interest.

**Acknowledgement**

This work was supported by ICIMOD's Cryosphere Initiative funded by Norway, and by core funds of ICIMOD contributed
by the governments of Afghanistan, Australia, Austria, Bangladesh, Bhutan, China, India, Myanmar, Nepal, Norway, Pakistan,
Sweden, and Switzerland. The views and interpretations in this publication are those of the authors and are not necessarily
attributable to ICIMOD.

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

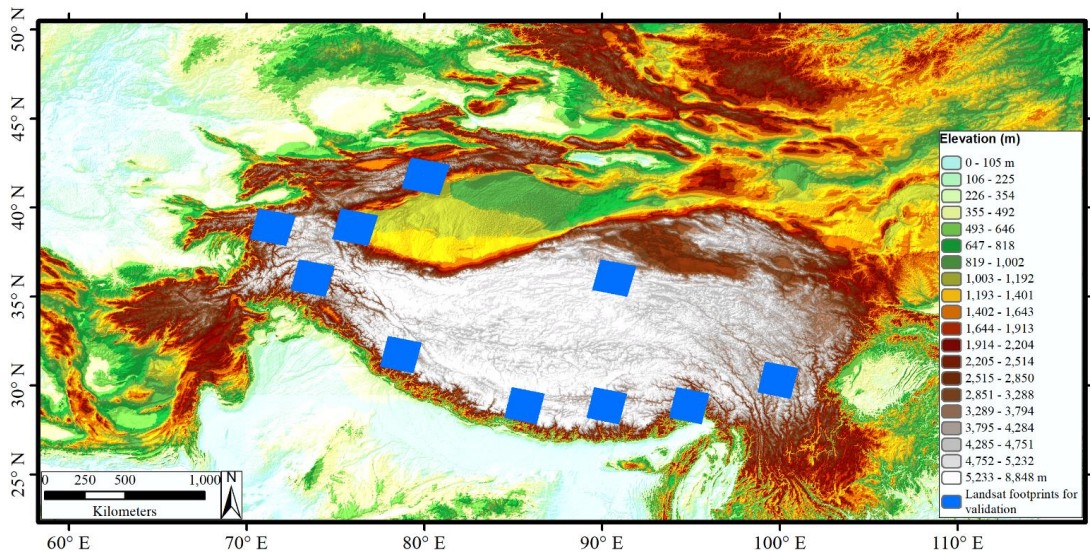

**Figure 1:** Study area map showing elevation throughout the region and Landsat 8 satellite scenes used for MODIS snow validation. Two images of each Landsat footprints shown in this map were used for validation.

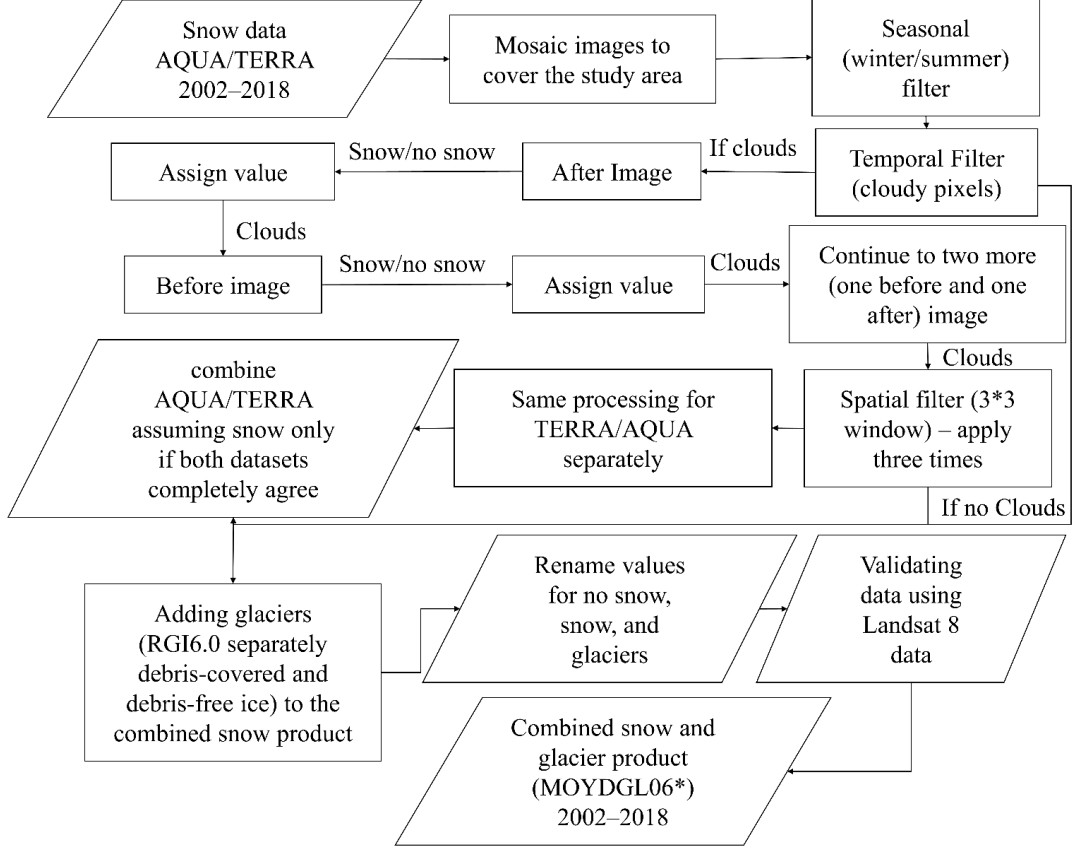

**Figure 2:** Methodology Flowchart.



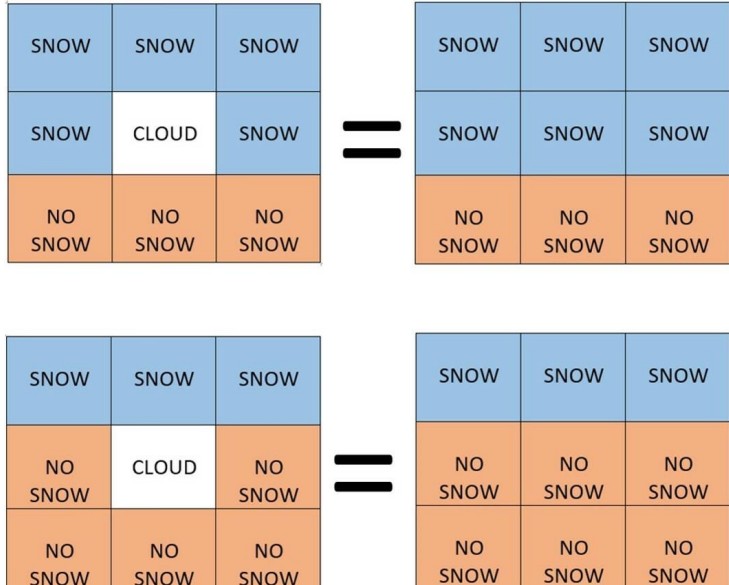

Figure 3: Spatial filter of the methodology describing cloudy pixels conversion to snow and no snow. If any of the surrounding majority pixels are snow or no snow, the cloudy pixels are assigned the same value, respectively.

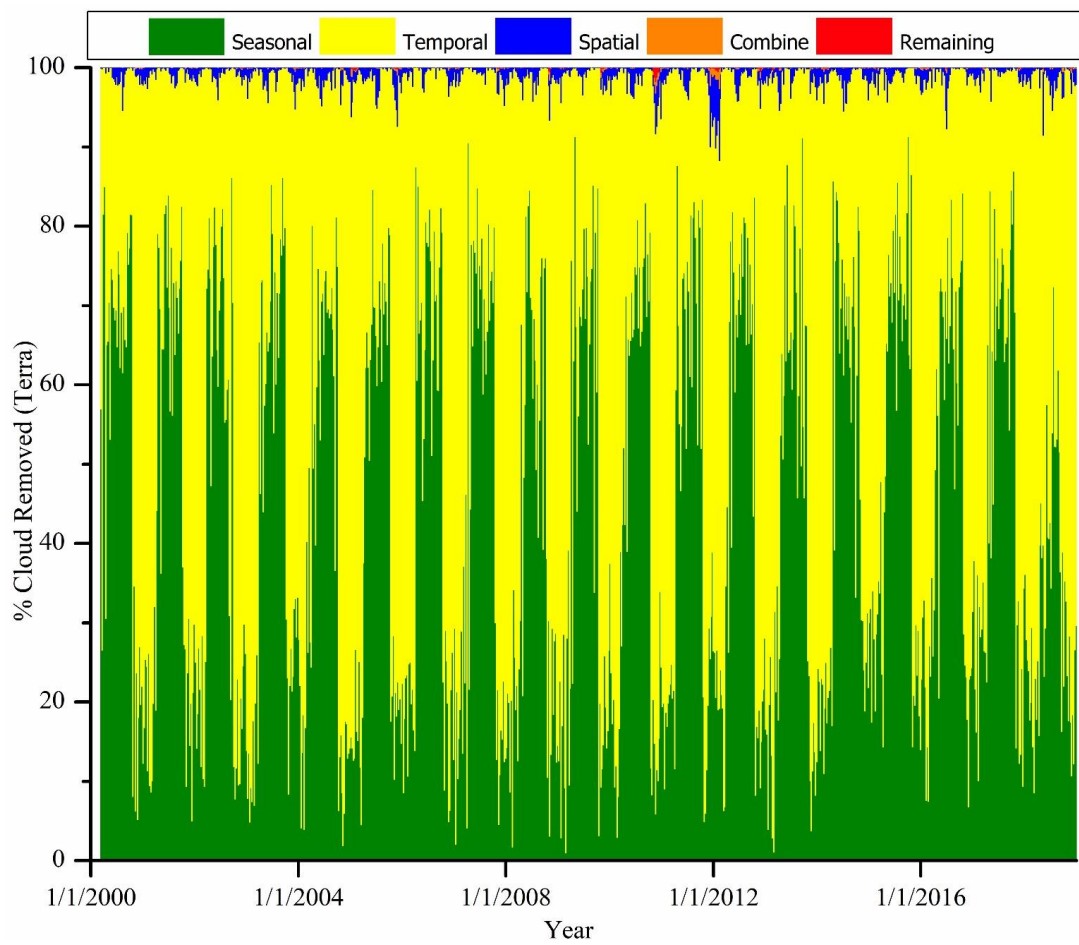

Figure 4: Cloud cover removed from the Terra product by extent, temporal filter, spatial filter, the combination of Terra and Aqua, remaining cloud cover.

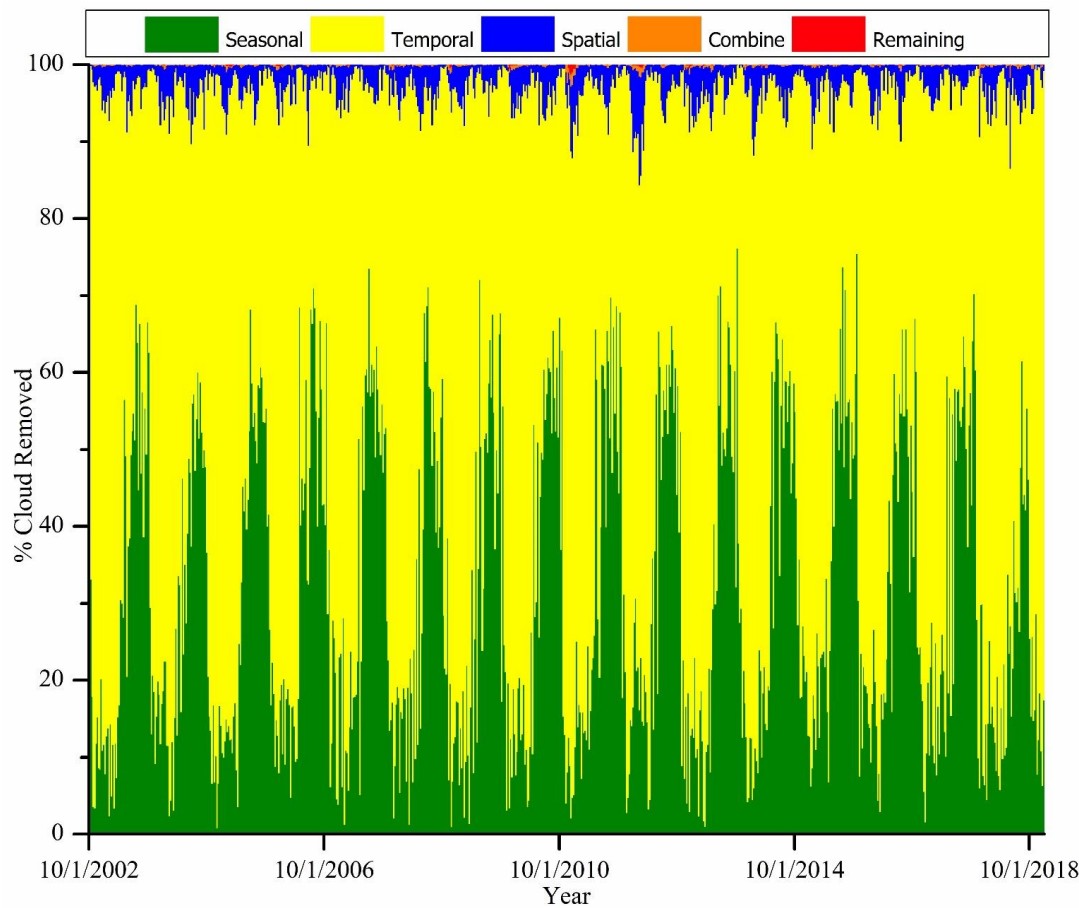

Figure 5: Cloud cover removed from the Aqua product by extent, temporal filter, spatial filter, the combination of Terra and Aqua, remaining cloud cover.

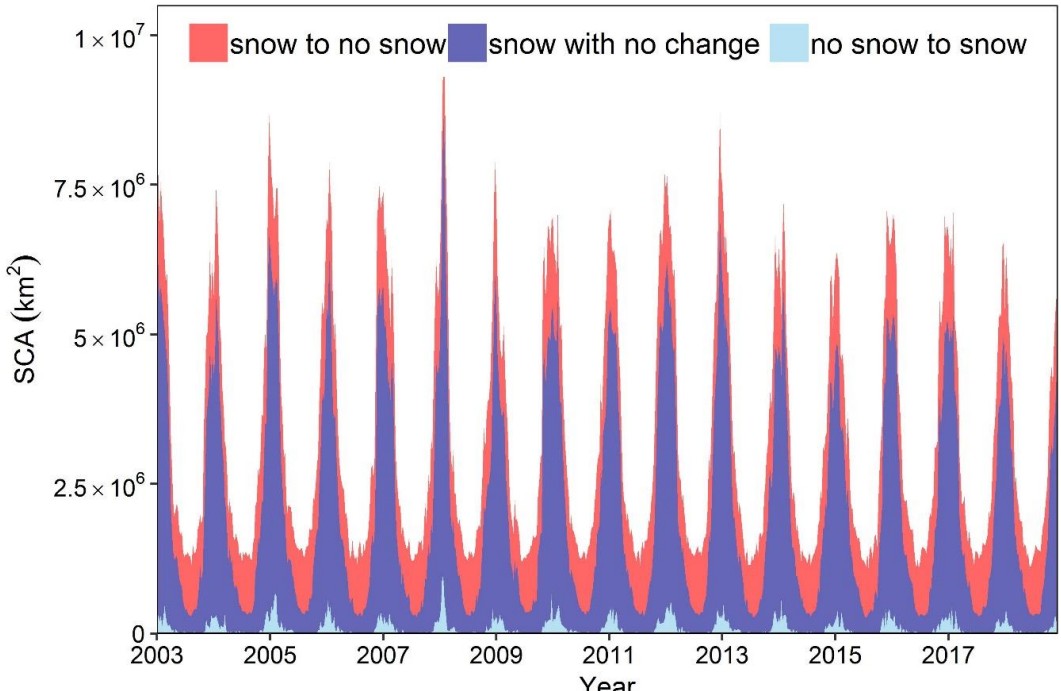

Figure 6: Improved combined snow and glacier product for the period between 2002 and 2018. The values -200 is the snow converted to no snow in the final product, 0 is no now either in the raw data or converted from cloudy pixels, 200 is snow without any change in the raw and final product, 210 is the no snow converted to snow mainly due to cloud cover.

Table 1: Validation of snow cover (peak snow cover period) for ten selected well-distributed areas to represent the study area derived by the original Terra MODIS (MOD10A2.006*), Aqua MODIS (MYD10A2.006*), and their separate and combined improved data.

| Landsat Path/Row | Landsat data acquisition date | cloud cover% | MODIS data acquisition date | accuracy raw Terra | accuracy raw Aqua | accuracy final Terra | accuracy final Aqua | Accuracy combined |
|---|---|---|---|---|---|---|---|---|
| 141/40 | 01/16/2018 | 0.90 | 2018009 | 0.87 | 0.84 | 0.87 | 0.84 | 0.86 |
| 132/39 | 11/01/2018 | 2.31 | 2018305 | 0.80 | 0.80 | 0.80 | 0.80 | 0.85 |
| 135/40 | 01/22/2018 | 2.69 | 2018017 | 0.90 | 0.87 | 0.72 | 0.87 | 0.91 |
| 138/40 | 12/29/2018 | 4.50 | 2018361 | 0.62 | 0.64 | 0.64 | 0.65 | 0.67 |
| 139/35 | 11/18/2018 | 4.61 | 2018321 | 0.69 | 0.66 | 0.76 | 0.73 | 0.79 |
| 150/35 | 12/17/2018 | 3.51 | 2018345 | 0.64 | 0.63 | 0.63 | 0.62 | 0.67 |
| 146/38 | 12/21/2018 | 2.87 | 2018353 | 0.74 | 0.71 | 0.77 | 0.74 | 0.79 |
| 147/31 | 11/26/2018 | 2.48 | 2018329 | 0.78 | 0.75 | 0.80 | 0.79 | 0.82 |
| 149/33 | 02/09/2018 | 2.98 | 2018033 | 0.67 | 0.63 | 0.71 | 0.68 | 0.70 |
| 152/33 | 12/15/2018 | 0.45 | 2018345 | 0.58 | 0.60 | 0.57 | 0.58 | 0.59 |



Table 2: Validation of snow cover (minimum snow cover period) for ten selected well-distributed areas to represent the study area derived by the original Terra MODIS (MOD10A2.006*), Aqua MODIS (MYD10A2.006*), and their separate and combined improved data.

| Landsat Path/Row | Landsat data acquisition date | cloud cover% | MODIS data acquisition date | accuracy raw Terra | accuracy raw Aqua | accuracy final Terra | accuracy final Aqua | Accuracy combined |
|---|---|---|---|---|---|---|---|---|
| 141/40 | 04/22/2018 | 7.24 | 2018105 | 0.89 | 0.79 | 0.89 | 0.85 | 0.90 |
| 132/39 | 01/01/2018 | 1.82 | 2018001 | 0.69 | 0.70 | 0.69 | 0.73 | 0.74 |
| 135/40 | 11/22/2018 | 3.16 | 2018321 | 0.89 | 0.84 | 0.88 | 0.84 | 0.91 |
| 138/40 | 01/11/2018 | 0.72 | 2018009 | 0.92 | 0.90 | 0.92 | 0.90 | 0.92 |
| 139/35 | 12/04/2018 | 3.10 | 2018337 | 0.67 | 0.64 | 0.70 | 0.70 | 0.75 |
| 150/35 | 09/12/2018 | 1.41 | 2018249 | 0.91 | 0.79 | 0.91 | 0.79 | 0.92 |
| 146/38 | 09/16/2018 | 3.28 | 2018257 | 0.92 | 0.83 | 0.92 | 0.83 | 0.93 |
| 147/31 | 01/10/2018 | 2.95 | 2018009 | 0.70 | 0.65 | 0.74 | 0.73 | 0.77 |
| 149/33 | 08/20/2018 | 2.33 | 2018225 | 0.97 | 0.92 | 0.97 | 0.93 | 0.97 |
| 152/33 | 09/10/2018 | 4.32 | 2018249 | 0.89 | 0.83 | 0.89 | 0.83 | 0.90 |

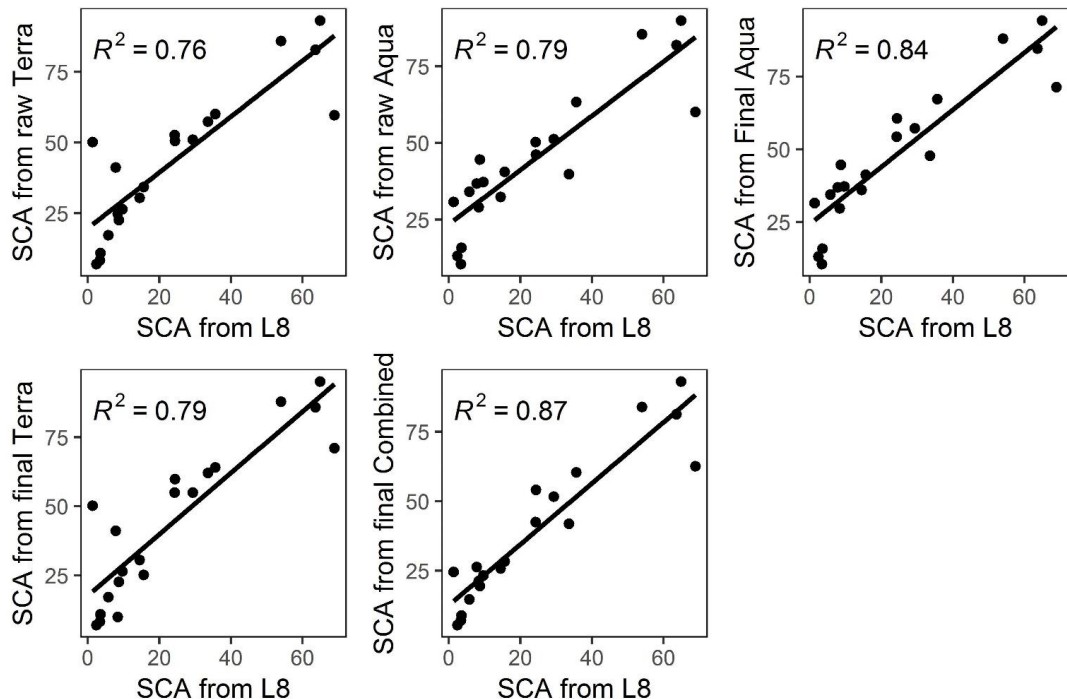

Figure 7: Correlation of the raw, and improved Terra/Aqua, and combined Terra/Aqua snow data with the Landsat 8 data. The description
5   SCA is snow cover area and L8 Landsat 8.

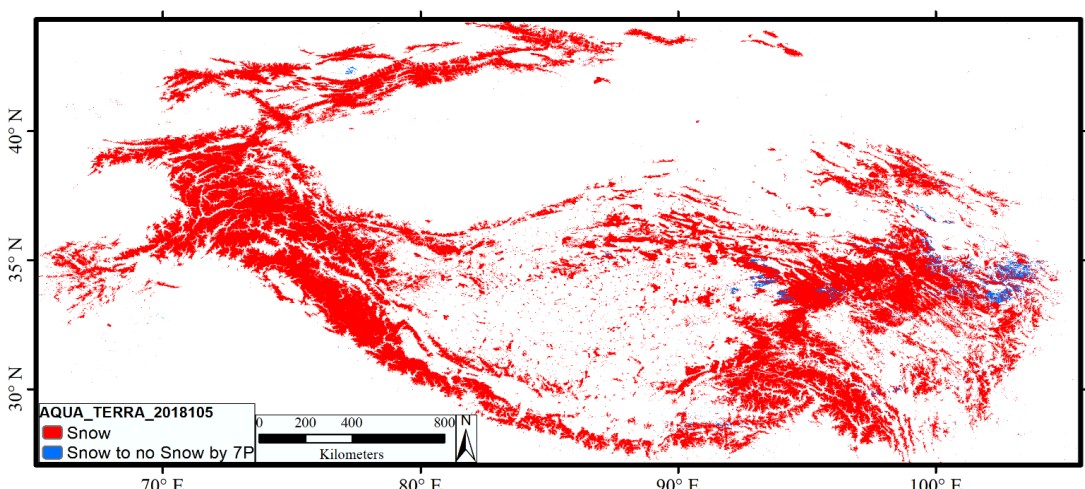

Figure 8: Snow underestimation by 7% persistency threshold in the combined Terra and Aqua snow product. The red colour is the snow with no changes whereas, blue is the underestimated snow by the persistence threshold.

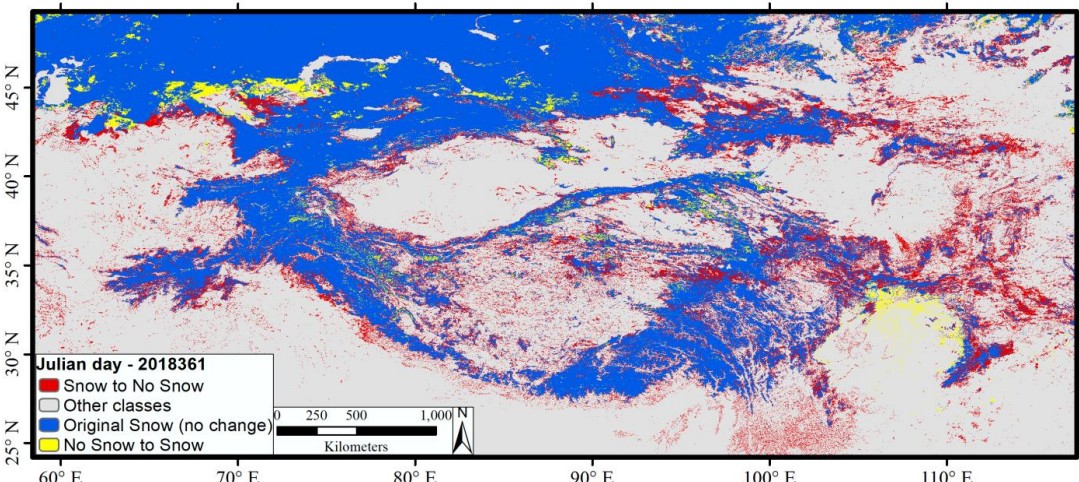

5    Figure 9: An example of the improved snow product showing the snow with no change, snow converted to no snow and no snow to snow by our methodology in the combined Terra and Aqua product. The blue and red is our final snow.

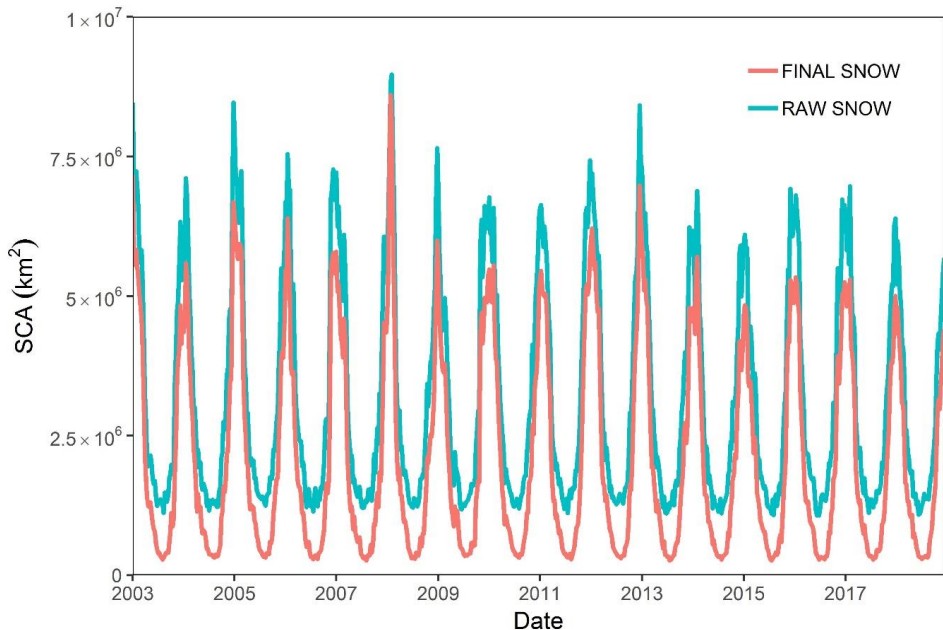

Figure 10: Raw and Final snow cover time series between 2002 and 2018 for the whole study area.