# Peer review of "A combined Terra/Aqua MODIS snow-cover and RGI6.0 glacier product (MOYDGL06\*) for the High Mountain Asia between 2002 and 2018"

_Earth System Science Data, 2019_

## Short Comment (SC1) · 21 Jun 2019

This paper generates snow data derived from MODIS onboard TERRA and AQUA. The paper combines MODIS TERRA and AQUA satellites snow data to reduce uncertainty/bias. The paper uses state of the art technology to generate a new snow dataset for the High Mountain Asia covering the period from 2003 to 2018. The data generation is well presented and the method is stepwise explained. The output is a complete product showing any changes in the original snow product, which is very useful for users. The data has a wide range of applications including hydrology, climate change, hydroglaciology, and modelling. I have few minor comments for the authors to address in order to improve the readability of the paper. 1. As Per the definition, maximum snow product has the tendency to overestimate snow. If there is a short term snow cover in lower elevation where the snow is not stable over time, the maximum approach results in more snow than in reality. It is important to know why 8-day composite data is used and why the authors prefer this product than the daily products? 2. There is a short temporal difference between both MODIS and Landsat, how did you manage to compare one single Landsat data set with an 8-day maximum composite and Why did you resample MODIS to Landsat-pixel size and not vice versa? 3. How many days temporal filter is applied in this study, it is unclear. 4. Combining Aqua and Terra —> This might be correct. But the retrieval accuracy changes due to different illumination conditions between Terra and Aqua. It has to be shown in detail, that the snow product (daily basis) between Terra and Aqua is more or less identical. Especially in rough topography there is a difference between both snow products. 5. Equations 4 and 5 seems identical, what exactly is the difference?

---

## Short Comment (SC2) · 26 Jun 2019

Li, X., Jing, Y., Shen, H. and Zhang, L.: The recent developments in spatio-temporally continuous snow cover product generation, Hydrol. Earth Syst. Sci. Discuss., (February), 1–28, doi:10.5194/hess-2018-633, 2019a.

The above reference should be updated, because it has been accepted and published by HESS.

Li, X., Jing, Y., Shen, H. and Zhang, L., 2019.  The recent developments in cloud

removal approaches of MODIS snow cover product. Hydrology and Earth System Sciences, 23(5): 2401-2416.

---

## Referee Comment (RC1) · Anonymous Referee #1 · 22 Jul 2019

Abstract: In your abstract you had mentioned and put a link to access the data that you produced (MOYDGL06\*) with a reference. You should remove it from here and you better put it in the data analysis section or in any other appropriate sections. Introduction: In your introduction please try to include the objective of the study and how your study will contribute in filling the existing gaps. The paper miss this point. I think you can also merge Section 3.6 and Section 5 together and you can remove section 5. General Comments and Questions: 1. Please define variables and symbols in your equations. 2. To what extent were your original MODIS Terra and Aqua data were disturbed by

cloud cover? 3. Can you show us the comparison map of pixels affected by cloud and the improved map after you are applying the three filtering techniques on mapping the snow cover? Your manuscript miss this important aspect. 4. How do you separate the debris-covered and debris-free glacier ice? 5. Do you have any reason why you choose Landsat 8 data for validation only in 2018? Why not in any other years of the study period? 6. Why the minimum and maximum snow periods are selected in 2018 only? 7. Have you tried to improve the existing snow detection algorithms to avoid overestimation of the MODIS snow cover data or you simply used the one developed by others from the literatures? Please try to discuss everything clearly. 8. Which snow detection threshold method have used in this study? Please mention it clearly. 9. "The seasonal filter removed approximately 44.66 % and 31.29 %, temporal filter removed 54.08 % and 65.48 %, spatial filter removed 99.91 % and 99.84 % of the total cloud cover existing mainly outside the snow cover extent in Terra and Aqua products." Line 32- 38. So, why temporal filter is the most effective step in cloud removal than the others? It is not clear for me. 10. Why the overall snow extent is showing significantly decreasing trend since 2013 as compared to the whole observation period between 2002 and 2018? Please elaborate this. 11. On Figure 6 caption please change the word "now" to "snow". 12. On Figure7, the unit for SCA is not mentioned. 13. From Figure 9, generally we can say that the number of pixels changed from snow to no snow are much more higher than the pixels changed from no snow to snow which shows that the uncertainty in snow cover underestimation due to cloud cover is less than that of the MODIS large swath width and poor spatial resolution. It also shows to use better spatial resolution snow cover product than the one you proposed. Please try to elaborate and discuss this point in your discussion part.

---

## Referee Comment (RC2) · Anonymous Referee #2 · 6 Nov 2019

Apologies to authors for this late review!

MODIS snow cover products are very important for many research and operational applications and this paper generates and describes a new product for High Mountain Asia, this is certainly a useful contribution to the community and I recommend publication subject to several mainly minor comments.

GENERAL COMMENTS

1. fSCA or NDSI in V6 is a useful measure of subpixel snow cover, especially if the data

is to to be assimilated in a modelling scheme to, for example simulate SWE. However this is not available in this study as you use the 8-day product. Can you explain this choice in more detail in the introduction please.

2. Related to point 1, as you implement filtering routines in this study, why did you choose to base the study on the 8day product (a strategy to minimise cloud cover) and not work with daily data? You may have had better temporal coverage in resulting dataset if you had done that? Please justify this decision in the text.

3. While in general, the paper is well written there are reasonably frequent slips in style and grammar, which would likely be caught by thorough reread.

4. Will this dataset be updated in time (annually or so?). Perhaps mention this in the conclusion if so.

5. While as a data paper methods are not expected to be novel, it would be good to emphasis more strongly in the introduction what the contribution of the paper is in terms of both a product that does not yet exist (spatial coverage) and any methodological developments.

6. Please include a README.txt in the data folder on Pangea that contains all meta-data required to use the data eg. codes etc.

SPECIFIC COMMENTS

p1l26 in –>during

p2l16 -> what is meant by 'improve the snow cover extent?'

p2l21 what is meant by 'somehow improved the quality of snowcover'

p2l23 suggest enormous -> large

p3l18 what is a year? calender? or hydrological?

p3l18 consider rephrasing "total seasonal snow cover extent was....." for clarity

p3l33 "then we go for the spatial filter " -poor language

p3l6 "Eq.(1)  only"

p3l35 when is melting expected to be negligible? Is this a fair assumption? Please expand.

p4l33 "differentiate <them>,"

p7l28"we conclude that clouds are not the main obstacle.." - can you expand on what is the main obstacle?

Fig2 'if no clouds' should be where division occurs at the 'Temporal filter' box.

Fig 4+5 could perhaps be a single 2-panel plot to save space as they are closely related.

Fig 6 looks like the plot starts in 2003, not 2002 as stated in the caption.

Table 1 inconsistent capitalisation

Table 2 inconsistent capitalisation

Fig 7. non-intuitive order. Should probably be three column plot 'terra', 'aqua', 'combined'. Is there no 'raw combined'?

Fig 9 caption says 'the blue and red is our final snow' - what does this mean? the legend says that red is snow converted to no snow. These statements seem inconsistent.

Fig 10, It seems that the large systematic shift in summer is at least partly due to 'snow' in the original dataset being reclassed as glacier ice. Does this play a role? If so please discuss that a bit more. The main explanation given is that falsely classified clouds are removed.

Can you make the coverage field on Pangea a polygon and not just a point - this would be more useful.

---

## Author Response (AR1)

**Responses to Anonymous Referee #1 Comments**

We would like to thank the reviewers for their valuable time in reviewing the manuscript and providing suggestions for improvement. We appreciate their feedback and constructive criticism. We would also like to thank the editor for giving us the opportunity to improve the manuscript.

Responses to the reviewer comments are specified below. Reviewer's comments are stated in black whereas the response is in blue color.

**Reviewer#1**

Abstract:

In your abstract you had mentioned and put a link to access the data that you produced (MOYDGL06*) with a reference. You should remove it from here and you better put it in the data analysis section or in any other appropriate sections.

Response: A link to access the data with a reference in the abstract is given following the journal format/recommendations. We are just following the journal format.

Introduction:

In your introduction please try to include the objective of the study and how your study will contribute in filling the existing gaps. The paper miss this point.

Response: The third paragraph of the Introduction highlights the issues/uncertainties in MODIS snow which are reduced in this study. In the second last sentence of the Introduction sentence which is now modified to make it more clearly, we have described the aim/objective of the study. It is now stated that "The aim of this study is to reduce uncertainty in MODIS snow data caused either by cloud cover or sensor's limitations, using a multi-step approach to removes cloud persistence causing underestimation and reduces an enormous amount of overestimation in snow cover mainly by the larger SZA of MODIS".

I think you can also merge Section 3.6 and Section 5 together and you can remove section 5.

Response: We agree and thank the reviewer for the suggestion. Sections 3.6 and section 5 are now combined.

**General Comments and Questions:**

1. Please define variables and symbols in your equations.

Response: We thank the reviewer for identifying this. All the variables and symbols in the equations are now defined and added to the revised manuscript. On page 4: line 7 we added "where S represents matrix, c denotes cloud, x and y are row and column index of S, t is time index."

2. To what extent were your original MODIS Terra and Aqua data were disturbed by cloud cover?

Response: Cloud cover affected TERRA and AQUA data by 5.31% and 6.52% on average. This is now added to the text in the manuscript (Page 6: Line 5-6).

3. Can you show us the comparison map of pixels affected by cloud and the improved map after you are applying the three filtering techniques on mapping the snow cover? Your manuscript miss this important aspect.

Response: We are thankful for the reviewer suggestion to make a new map and show the cloudy pixels in raw data converted to snow and no snow. We have added Figure 4 showing improved map with cloudy pixels converted to snow and no snow by temporal and spatial filters.

4. How do you separate the debris-covered and debris-free glacier ice?

Response: Debris-cover and debris-free glacier ice are available at RGI database (as mentioned on Page 4: line 37-38. We included/merged the available glacier data (debris cover and debris free) to our snow product as described on Page 5: Line 1-5. "*In the regions where snow and glaciers both exist, it is challenging to differentiate particularly in the accumulation period. Also, the glacier ice mainly in the late ablation season is difficult to map using the MODIS algorithm for snow detection when the albedo of the glacier surface is comparatively low. MODIS is incapable of mapping ice under the debris. Therefore, we used the latest Randolph Glacier Inventory version 6.0 (RGI6.0) (RGI Consortium, 2017), partly developed by Mölg et al., (2018) and supraglacial debris cover for RGI 6.0 by Scherler et al. (2018), resampled into the MODIS pixel size and merged it into the combined MODIS data. A combined snow and glacier cover (debris-covered and debris-free) product was developed which will be useful mainly for glacio-hydrological applications*". This kind of dataset is very demanding for cryospheric research in particular glacio-hydrological modelling. We believe that our dataset will add significant role in hydrological modelling.

5. Do you have any reason why you choose Landsat 8 data for validation only in 2018? Why not in any other years of the study period? 6. Why the minimum and maximum snow periods are selected in 2018 only?

Response: We selected 2018 randomly to compare our results with the latest observation period. The study area is quite large, therefore, we selected 10 locations as a representative of the whole region and validation of snow data. We divided the year into winter and summer and validated the snow to make the validation seasonally well distributed. We have selected 20 Landsat images for validation which is quite extensive for validation. Our validation captures the variability and uncertainty quite well.

7. Have you tried to improve the existing snow detection algorithms to avoid overestimation of the MODIS snow cover data or you simply used the one developed by others from the literatures? Please try to discuss everything clearly.

Response: As the snow overestimation is not because of the snow detection algorithms (Page 2: Line 11-14). The reason behind snow overestimation is " Larger Sensor Zenith Angle (SZA) (Li et al., 2016) and low spatial resolution (Hou et al., 2019; Huang et al., 2017) mainly causes overestimation of snow. The overestimation is also significantly influenced by the broad swath of MODIS that amplifies the edge-pixels more than four times compared to the pixels at the image centre (Zeng et al., 2011; Zhang et al.,

2017)". "We used 8-day maximum snow extent product version 6 of the MODIS onboard Terra (MOD10A2.006*) and Aqua (MYD10A2.006*) available from February 2000 and July 2002, respectively with 500 m spatial resolution for the Hindukush, Karakoram, and Himalaya (HKH) and surroundings. This version minimises the error of omission and commission compared to version 5 primarily in clear sky conditions as described by Riggs et al., (2016). In collection 6, band 6 of AQUA is restored instead of the previously used band 7 in calculating NDSI making the algorithm similar to that used for TERRA (Riggs et al., 2016) which helps to reduce an additional uncertainty in AQUA snow cover." Page 2: Line 35-41.

8. Which snow detection threshold method have used in this study? Please mention it clearly.

Response: We used Terra (MOD10A2.006*) and Aqua (MYD10A2.006*) products. These are level 2 products and give snow data represented as (200) as mentioned in Page 3 Line 2. We didn't derive snow from images but the data is already available as snow. We processed the available product and improved the final snow with a significant improvement of reducing underestimation of 3.66% on average because of cloud cover and 46% of overestimation because of MODIS sensor larger zenith angle.

9. "The seasonal filter removed approximately 44.66 % and 31.29 %, temporal filter removed 54.08 % and 65.48 %, spatial filter removed 99.91 % and 99.84 % of the total cloud cover existing mainly outside the snow cover extent in Terra and Aqua products." Line 32- 38. So, why temporal filter is the most effective step in cloud removal than the others? It is not clear for me.

Response: The temporal filter basically considers any pixels which are cloud-free in any of the consecutive 40 days rather than seasonal (consecutive cloud cover for the six months), and majority spatial filter (considering only surrounding 8 pixels). However, it is important to mention that all the filters including temporal filter only improve cloudy pixels (3.66% of the total snow) to snow or no snow.

10. Why the overall snow extent is showing significantly decreasing trend since 2013 as compared to the whole observation period between 2002 and 2018? Please elaborate this.

Response: Thank you for pointing out this question. Our aim and scope of the paper is to improve the snow data only and we do not work on the trend analysis. For trend analysis and the possible reasons of dynamics in snow cover dynamics, we are carrying out another study to analyze the snow cover in detail for different regions (Karakoram, Himalaya, Hindukush, and Tibetan Plateau).

On Figure 6 caption please change the word "now" to "snow".

Response: Thanks to the reviewer for pointing out the typo, "now" is corrected as "snow" in the Figure 6 caption.

On Figure 7, the unit for SCA is not mentioned.

Response: The unit (%) is now added to Figure 7, thanks to the reviewer for the comment. The caption is also slightly revised by removing second sentence as "Correlation of the snow cover area (SCA) from raw, improved Terra/Aqua, and combined Terra/Aqua snow data with the Landsat 8 (L8) data".

From Figure 9, generally we can say that the number of pixels changed from snow to no snow are much more higher than the pixels changed from no snow to snow which shows that the uncertainty in snow cover underestimation due to cloud cover is less than that of the MODIS large swath width and poor spatial resolution. It also shows to use better spatial resolution snow cover product than the one you proposed. Please try to elaborate and discuss this point in your discussion part.

Response: We completely agree to the reviewer comment and have therefore highlighted the overestimation by MODIS in this study. We have added some explanation on Page 6 in the Discussion section (as described in italic font below). The method of combining Terra and Aqua is also an inter-verification of the snow derived by both the satellites. Our results indicate that on average approximately 46% of the total snow on average is overestimated by MODIS. This significant difference in the snow data is mainly due to the large swath and low spatial resolution of MODIS which makes it challenging to map snow cover accurately, particularly at the edges of each image. Similarly, the off-nadir view makes the sensor zenith angle larger causing it to replicate the edge pixels. Whereas, the underestimation is mainly caused by the cloud cover but is insignificant, i.e. 3.66% of the snow on average. *These results suggest that the uncertainty of underestimation in the snow cover due to cloud cover is quite low (approximately 7% of the overall uncertainty), in contrast to the overestimation uncertainty contribution of about 93%. It is to be noted that this cloud cover is significantly reduced in the 8-day composite as the cloud cover is the least possible in consequent eight days.* We are more confident about the MODIS snow cover derived from our method. Combining the snow with the glacier cover (debris-covered and debris-free) makes it more comprehensive and usable for various hydro-glaciological applications. The glacier ice captured by MODIS as snow is represented as 200 (snow). We combined glaciers uncaptured as snow by MODIS in the combined product representing debris-covered and debris-free ice as 240 and 250, respectively. These values (240 and 250) may be ignored or converted to no snow if the user of the data is interested only in the MODIS snow product. In this case, the values 200 and 210 can be considered as the final snow.

Comparison of the snow cover area estimated by Landsat and MODIS Aqua/Terra raw/final and combined product shows that our methodology improved the accuracy by 10% from 77% to 87% on average reducing the inevitable overestimation for twenty well-distributed (in space and time) Landsat scenes. The remaining overestimation is constrained by low spatial resolution and large swath. *Therefore, for very small scale studies, low spatial resolution data, including our improved snow product is not recommended.*

**Referee #2**

MODIS snow cover products are very important for many research and operational applications and this paper generates and describes a new product for High Mountain Asia, this is certainly a useful contribution to the community and I recommend publication subject to several mainly minor comments.

We would like to thank the reviewer for the positive feedback and suggestions to improve the readability of the manuscript. We carefully revise our manuscript as per the recommendations of the reviewer. Responses to the reviewer comments are specified below. Reviewer's comments are stated in black whereas the response is in blue color.

GENERAL COMMENTS

1. fSCA or NDSI in V6 is a useful measure of subpixel snow cover, especially if the data is to be assimilated in a modelling scheme to, for example simulate SWE. However this is not available in this study as you use the 8-day product. Can you explain this choice in more detail in the introduction please.

Response: We thank the reviewer for the comment. The criteria of selecting 8-day snow product is now explained in the last paragraph of the introduction as "The aim of this study is to reduce uncertainty in MODIS snow data caused either by cloud cover (underestimation) or limitations due to large SZA (overestimation), using a multi-step approach. The daily binary and fractional products are useful for simulation and modelling of the cryosphere and hydrology but the use of existing products may lead to significant uncertainty in the results due to above limitations. Therefore, we improved the 8-day composite products in which not only the cloud cover is minimized but the combination of TERRA and AQUA reduces the overestimation of snow due to large SZA. A long-term (2002-2018) meticulous estimate of the combined TERRA and AQUA 8-day composite snow cover for the HMA (Fig. 1) will facilitate climate, glacio-hydrological modelling, understanding the present dynamics of the cryosphere in the region (Brun et al., 2017; Muhammad et al., 2019a). The product will also lead to improve and develop associated products, e.g., daily snow water equivalent, fractional snow cover, and daily binary snow data (Alonso-González et al., 2018; Painter et al., 2016)."

2. Related to point 1, as you implement filtering routines in this study, why did you choose to base the study on the 8day product (a strategy to minimise cloud cover) and not work with daily data? You may have had better temporal coverage in resulting dataset if you had done that? Please justify this decision in the text.

Response: We thank the reviewer for the comment and clarification. We have now explained our criteria of selecting 8-day composite product as "The temporal and spatial filters can be efficient for the daily products but the uncertainty due to larger SZA cannot be reduced. The daily binary and fractional products are useful for simulation and modelling of the cryosphere and hydrology but the use of existing products may lead to significant uncertainty in the results due to above limitations. Therefore, we improved the 8-day composite products in which not only the cloud cover is minimized but the combination of TERRA and AQUA reduces the overestimation of snow due to large SZA."

3. While in general, the paper is well written there are reasonably frequent slips in style and grammar, which would likely be caught by thorough reread.

Response: We have now revised the manuscript for possible grammatical mistakes and corrected accordingly. We hope the revision will be satisfactory for the editor and reviewers. We will copy edit the manuscript through copy editor if the editor asks to do so.

4. Will this dataset be updated in time (annually or so?). Perhaps mention this in the conclusion if so.

Response: It is a one time and long-term data improved dataset. We do not promise any updates instead provide the R code for processing any additional data if required.

5. While as a data paper methods are not expected to be novel, it would be good to emphasis more strongly in the introduction what the contribution of the paper is in terms of both a product that does not yet exist (spatial coverage) and any methodological developments.

Response: We have now revised the introduction section of the manuscript. In the last paragraph of the introduction section, the possible applications, improvement, selection criteria, and rationale behind the selection of data is added in the revised manuscript.

6. Please include a README.txt in the data folder on Pangea that contains all metadata required to use the data eg. codes etc.

Response: The README (instruction.txt) and R code files are available as a supplement to the manuscript file. In Pangaea, it is also described that the data is associated to the ESSD manuscript. If the editor suggest to add these files to the Pangaea, we will do so.

p1l26 in –>during

Revised "in" as "during"

p2l16 -> what is meant by 'improve the snow cover extent?'

revised as "improve the snow cover data"

p2l21 what is meant by 'somehow improved the quality of snowcover'

There were some improvement in the data but the snow data still contain error of omission and commission. This study significantly improved the data.

p2l23 suggest enormous -> large

The statement is revised and the sentence is removed containing enormous.

p3l18 what is a year? calender? or hydrological?

"hydrological" added

p3l18 consider rephrasing "total seasonal snow cover extent was....." for clarity

revised as "maximum seasonal accumulated snow extent"

p3l33 "then we go for the spatial filter " -poor language

revised as "we stop temporal filter and instead use spatial filter for removing the remaining cloudy pixels."

p3l6 "Eq.(1)  only"

added by in between Eq.(1) and only

p3l35 when is melting expected to be negligible? Is this a fair assumption? Please expand.

Under cloudy conditions as stated in the previous sentence. Under the cloud cover conditions, the melting is significantly low and negligible as compared to clear sky conditions.

p4l33 "differentiate <them>,"

"them" added after "differentiate"

p7l28"we conclude that clouds are not the main obstacle.." - can you expand on what is the main obstacle?

The large SZA caused an overestimation of 46% in snow, which is the main obstacle. It is already explained in the text.

Fig2 'if no clouds' should be where division occurs at the 'Temporal filter' box.

If no clouds, then we go combine TERRA and AQUA step. "if no clouds" is at correct place.

Fig 4+5 could perhaps be a single 2-panel plot to save space as they are closely related.

These figures become too congested when combined, therefore, we keep it separated.

Fig 6 looks like the plot starts in 2003, not 2002 as stated in the caption.

The plot starts from the date as mentioned (October 1).

Table 1 inconsistent capitalisation

There is no inconsistency

Table 2 inconsistent capitalisation

There is no inconsistency

Fig 7. non-intuitive order. Should probably be three column plot 'terra', 'aqua', 'combined'.

Is there no 'raw combined'?

The figure is made to make it easily visualize by keeping Terra and Aqua raw and final besides each other.

Fig 9 caption says 'the blue and red is our final snow' - what does this mean? the legend says that red is snow converted to no snow. These statements seem inconsistent.

We thank the reviewer for the comment and correcting us. Actually, the blue and yellow are our final snow, the caption is revised and corrected.

Fig 10, It seems that the large systematic shift in summer is at least partly due to 'snow' in the original dataset being reclassed as glacier ice. Does this play a role? If so please discuss that a bit more. The main explanation given is that falsely classified clouds are removed.

The snow in both the summer and winter seasons are overestimated by MODIS which is improved by our methodology. As our product is a combined snow and glacier product, the mixing of snow and ice may have a minute effect.

Can you make the coverage field on Pangea a polygon and not just a point - this would be more useful.

There is no option to add a polygon of our study area. Actually, the selection of coverage field is controlled by Pangaea administrator and we do not have any control on it. We thank the reviewer for the constructive feedback on the manuscript.

**Short Comment SC1**

This paper generates snow data derived from MODIS onboard TERRA and AQUA. The paper combines MODIS TERRA and AQUA satellites snow data to reduce uncertainty/bias. The paper uses state of the art technology to generate a new snow dataset for the High Mountain Asia covering the period from 2003 to 2018. The data generation is well presented and the method is stepwise explained. The output is a complete product showing any changes in the original snow product, which is very useful for users. The data has a wide range of applications including hydrology, climate change, hydro-glaciology, and modelling. I have few minor comments for the authors to address in order to improve the readability of the paper.

Response: We are thankful to the reviewer for the constructive review and comments. We carefully considers all the comments and revise the paper accordingly. Our point by point response is given in blue color whereas, the comments are in black.

1.      As Per the definition, maximum snow product has the tendency to overestimate snow. If there is a short term snow cover in lower elevation where the snow is not stable over time, the maximum approach results in more snow than in reality. It is important to know why 8-day composite data is used and why the authors prefer this product than the daily products?

Response: We agree to the reviewer that the 8-day composite may overestimate snow. However, the main constrain in retrieving daily snow is the cloud cover. Even after using 8-day composite data (which is affected by clouds if it is persistent continuously for 8 consecutive days), there were clouds over 3.66% of the study area on average in the observation period. In addition, the large sensor azimuth angle (SZA) produce uncertainty (overestimation or underestimation) and makes the daily product significantly uncertain. We not only reduced the underestimation by removal of cloud cover but also reduced underestimation of uncaptured snow in one or more days of the 8 consecutive days and removed significantly large amount of overestimation (46% of the original snow) also caused by SZA.

2. There is a short temporal difference between both MODIS and Landsat, how did you manage to compare one single Landsat data set with an 8-day maximum composite and Why did you resample MODIS to Landsat-pixel size and not vice versa?

Response: We assume that the snow cover change is insignificant in each 8-days composite of MODIS. Although, the snow changes continuously but the snow cover variation is insignificant as compared to the uncertainty in snow extent from the large pixel size (500 m) of MODIS. We resampled MODIS to avoid data loss as resampling high spatial resolution to low resolution is susceptible to data loss because multiple pixels (consist of both snow and no snow) are converted to one pixel (either snow or no snow).

3. How many days temporal filter is applied in this study, it is unclear.

Response: Temporal filter considers two before and two after 8-days composite images for the cloudy pixels in the observed 8-day image, this means that for the cloudy pixels the longest days is 40.

4. Combining Aqua and Terra —> This might be correct. But the retrieval accuracy changes due to different illumination conditions between Terra and Aqua. It has to be shown in detail, that the snow product (daily basis) between Terra and Aqua is more or less identical. Especially in rough topography there is a difference between both snow products.

Response: We agree that the retrieval accuracy may change but this may affect daily snow retrieval. As we use 8-day composite, the accuracy of snow retrieval is significantly improved as we consider snow if both the products retrieve pixels as snow. We do not agree that the both Terra and Aqua be identical because the sensor zenith angle is the main factor to cause overestimation. Our results of 46% overestimation is a clear example of uncertainty in the rough topography for both the products.

5. Equations 4 and 5 seems identical, what exactly is the difference?

Response: These equations seem identical but are different. In equation 4, the pixels is snow if Terra is snow or cloud and Aqua is snow, WHEREAS, in equation 5, the pixels is snow if Aqua is snow or cloud and Terra is snow.

**Short Comment SC2**

Li, X., Jing, Y., Shen, H. and Zhang, L.: The recent developments in spatio-temporally continuous snow cover product generation, Hydrol. Earth Syst. Sci. Discuss., (February), 1–28, doi:10.5194/hess-2018-633, 2019a. The above reference should be updated, because it has been accepted and published by HESS. Li, X., Jing, Y., Shen, H. and Zhang, L., 2019. The recent developments in cloud removal approaches of MODIS snow cover product. Hydrology and Earth System Sciences, 23(5): 2401-2416.

Response: We thank the author of above paper for updating us about the final version publication. The reference is updated as above.

[revised manuscript text omitted]

---

## Author Response (AR2)

**Editor Comments and Response**

We are thankful to the editor for suggestions to clarify and improve the readability of our manuscript. We carefully consider all the suggestions and revise the manuscript accordingly. We do hope the revision will be satisfactory for the editor. We have thoroughly reviewed and revised the manuscript where necessary to avoid any misinterpretations.

i) please let check the language, there are still minor issues, e.g. missing articles
Response: The manuscript is thoroughly reviewed for any grammatical issues and corrected.
ii) please spell out all abbreviations, when abbreviations first appear, e.g. RGI6.0 glacier, NDSI
Response: All the abbreviations spell out on the first appearance.
iii) do not use the term 'raw' image or 'raw' data in text and figures as you refer already to highly developed product levels and not e.g. top-of-radiance data
Use instead official product names
Response: We thank the editor for the suggestion. We agree and revise raw as MOD10A2 and MYD10A2 Collection 6 (C6) throughout the manuscript.
Examples are p.13 § 3.3., last sentence 'In addition, figure 4 shows a raw image with cloudy pixels converted to snow and no snow by temporal and spatial filters.' -> e.g. use the official product name
§ 3.6 The combined product shows a significant improvement over the raw snow data -> e.g. use the official product name
§ 4 The overestimation in Terra and Aqua MODIS 8-day raw products 35 (MOD10A2*/MYD10A2*) is enormous -> e.g. omit the term 'raw'
etc please check throughout the manuscript text accordingly
Response: The raw replaced with MOD10A2.006 and MYD10A2.006 and partly as Collection 6 (C6) throughout the manuscript. We are thankful to the editor for the suggestion to improve the readability of our manuscript.
Please also change figure 4, figure 11 and caption, and column titles in table 2 accordingly,
Response: Figures 4, 8, and 11 and their captions revised. Also, the titles and caption of table 2 revised.
data publication
PANGAEA abstract
It is important that the PANGAEA data publication stands for it own with all information available.
i) please start the introduction with a sentence about the data set, geographic region and time period covered. e.g. The MOYDGL06* product MOYDGL06_2002_2018_HMA is an enhanced snow cover product covering the time period 2002 to 2018 with x temporal and x spatial resolution specifically developed for the .. region. Information that the code in the file name is the MODIS Julian day code. ..that the file format is geotiff
ii) please consider Referee #2's comment and offer readme file on the PANGAEA landing page of your data publication. This will be feasible if you address PANGAEA, and they will offer it as download possibility please ad a sentence at the end of your abstract that the readme contains all metadata required to use the data
Response: PANGAEA abstract is revised as suggested. "The data contains an enhanced MODIS 8-day Terra and Aqua snow-cover combined product merged with Randolph Glacier Inventory (RGI6.0). The input data used to generate this product are MOD10A2.006* and MYD10A2.006* representing Terra and Aqua MODIS 8-day composite collection 6 (C6) snow-cover, respectively. The data is specifically developed for the High Mountain Asia (HMA) with the geographic coverage between latitude 24.32 N− 49.19 N and Longitude 58.22 E − 122.48 E. The data MOYDGL06_2002_2018_HMA is an enhanced snow cover and glacier combined product covering the period between 2002 and 2018. The data is available with eight-day temporal resolution and 500 m spatial resolution. The name of the product is derived from MODIS Terra (MOD) MODIS Aqua (MYD), and Glacier (GL), Version 6 (06) as MOYDGL06*. The product is 8-day composite described in Julian day and each year has 46 eight-day composite images in GeoTIFF file format as described in the associated readme.TXT. The R code developed for this product is available at https://github.com/amrit-thapa-2044/essd_modis_paper. For more details about the data, please read the paper associated with this data at https://doi.org/10.5194/essd-2019-78."
Also, we have now added a readme.txt file as an attachment to the PANGAEA.
We have given a link to our ESSD manuscript, which we will update once we get the final acceptance from the editor.

software publication

ESSD encourages publication of data and tools and avoids supplements. Please publish your software, e.g. on github and show this information and link in the paper,
https://zonca.github.io/2017/02/publish-research-software-github.html

Response: The R code associated with a readme file is now published at GitHub: https://github.com/amrit-thapa-2044/essd_modis_paper

Details:

Abstract

L21 ad MODIS to Terra and Aqua

Response: MODIS added

Method

Titles 3.1 seasonal filter 3.2 temporal filtering 3.3. spatial filtering - please choose for consistency either filter or filtering

Response: revised seasonal filter -> seasonal filtering

L13 passive remote sensing data -> please ad 'optical' because of passive microwave remote sensing-> passive optical remote sensing data

Response: revised passive remote sensing data -> passive optical remote sensing data

p.13 L1 please change sentence, e.g.we change from the temporal filter to the spatial

Response: sentence revised as suggested.

3.4 title sounds unspecific, e.g could be more informative something in the line like 'merging Aqua and Terra filtered snow products'

Response: The title in 3.4 is revised as Combining Terra and Aqua filtered snow products

3.6. Landsat 8 images as ground truth -> as Landsat data are optical remote sensing data better the term ground truth is not used

better The final product was validated to assess the accuracy of the improved snow product using snow derived from Landsat 8 (United States Geological Survey, USGS) images for the year 2018 during both summer and winter seasons

Response: Thanks to the editor for the suggestion. We have now removed the term ground truth.

The snow was classified in Landsat following similar criteria applied for MODIS snow product, using NDSI based on Landsat [change to Landsat 8] bands 3 (0.53–0.59 µm) and 6 (1.57–1.65 25 µm)

Sentence and your technical processing remains unclear - 'followed by the reflection in near-infrared light greater than 11 % to prevent water from being incorrectly classified as snow'

Do you mean that you apply a threshold of 11 % Landsat 8 surface reflectance of masking out all reflectance values < 11 % before calculating the NSDI? Please extend on your method

Please use in this context the term reflectance, not reflection

Response: We thank the editor for suggestions to improve the readability and clarify the above statements. The revised statements are "The snow was classified in Landsat using the similar criteria which were applied for MODIS snow products, using NDSI based on Landsat 8 bands 3 (0.53–0.59 µm) and 6 (1.57–1.65 µm). Only those positive NDSI values/pixels are considered as snow having reflectance > 11 % in the near-infrared band. The reflectance threshold is to prevent water from being incorrectly classified as snow."

Also, the term reflection is revised as reflectance.

§ 6 Unclear sentence 'We conclude that clouds are not the main obstacle in the MODIS 8-day composite product as it reduces only 3.66% of snow.' Please change sentence into as the new product MOYDGL06* (? Is this the correct interpretation?) reduces only 3.66% of the snow –

Response: The above is in the original Terra and Aqua MODIS snow. Therefore, the revised statement is "We concluded that clouds are not the main obstacle in the MOD10A2 and MYD10A2 C6 products as it reduces only 3.66% of the snow."

Figure 6

Can you include more detail: e.g. either you could ad filter to seasonal, temporal and spatial inside the figure itself or provide it as an abreviation and describe these short codes in the figure caption. Use better the term merge than the term combine

Response: Description of the legends in Figure 5 and 6 are now added to the caption. We do hope, it is now clear to understand.

Figure 7

Include MOYDGL06* in figure caption

Response: MOYDGL06* added in the figure caption
Figure 10

Figure caption 'The blue and yellow is our final snow'. Please change this sentence e.g. blue and yellow colour codes are the xxx product

Response: Caption revised as suggested

[revised manuscript text omitted]